# Electrolyte Intake and Major Food Sources of Sodium, Potassium, Calcium and Magnesium among a Population in Western Austria

**DOI:** 10.3390/nu12071956

**Published:** 2020-06-30

**Authors:** Natalia Schiefermeier-Mach, Sabrina Egg, Judith Erler, Verena Hasenegger, Petra Rust, Jürgen König, Anna Elisabeth Purtscher

**Affiliations:** 1Department of Health Care and Nursing, FH Gesundheit, Health University of Applied Sciences Tyrol, 6020 Innsbruck, Austria; natalia.schiefermeier-mach@fhg-tirol.ac.at; 2Department of Dietetics, FH Gesundheit, Health University of Applied Sciences Tyrol, 6020 Innsbruck, Austria; sabrina.egg@fhg-tirol.ac.at (S.E.); judith.erler@fhg-tirol.ac.at (J.E.); 3Department of Nutritional Sciences, University of Vienna, 1090 Vienna, Austria; verena.hasenegger@univie.ac.at (V.H.); petra.rust@univie.ac.at (P.R.); juergen.koenig@univie.ac.at (J.K.)

**Keywords:** electrolytes, minerals, dietary intake, food sources, dietary patterns

## Abstract

Dietary intake of sodium, potassium, calcium and magnesium has a strong impact on personal health. In order to understand possible correlations with regional occurrence of diseases and to develop new dietary strategies, it is necessary to evaluate food choices in defined geographic areas. The aim of this study was to analyze daily consumption and major food sources of electrolytes with an emphasis on dietary patterns. In this representative, cross-sectional study, daily foods of 463 adults were assessed with two nonconsecutive 24-h recalls. Our results show high sodium and low potassium consumption in all age groups in both men and women. Furthermore, more than half of investigated persons had low calcium, and 40% indicated low dietary magnesium intake. Only 1% of our study population reached the recommended values for all electrolytes, while 13% consumed adequate levels of three electrolytes. Moreover, 14% did not reach reference levels for any of the four minerals. A further comparison of dietary patterns and food preferences showed significant differences in major food groups including nonalcoholic drinks, fruits, vegetables, legumes, milk products, vegetable oil, bread and sweets. Our results are important for further evaluations of nutrition intake and the development of new dietary strategies.

## 1. Introduction

Sufficient intake of vitamins and minerals, preferably as part of our daily food intake, is generally recommended to prevent chronic diseases and promote health. Sodium, potassium, calcium and magnesium, also known as electrolytes, have numerous biological effects in the human body, whereas their imbalances have a strong impact on personal health. Electrolyte balance is essential for bone composition and function, muscular physiology, neuronal processes, oxygen transport, acid-base homeostasis and many other biological processes. Even small deviations in blood concentrations of electrolytes can lead to serious health problems and even increased mortality [1].

Sodium (Na) is a cation in the extracellular fluid, which largely determines its volume and osmotic pressure. It is also present in the intracellular fluid, where it contributes to the cell membrane potential and enzymatic reactions. Together with chloride, dietary sodium is mostly consumed with table salt (NaCl) [2,3]. Potassium (K) is the most common cation of the intracellular fluid, where it is important for the regulation of membrane potential, maintenance of osmotic pressure, electrolyte homeostasis and acid-base-balance. Potassium also functions in controlling nerve impulses, protein biosynthesis and the conversion of blood sugar into glycogen. Although extracellular potassium only accounts for 2%, both increases and decreases of extracellular potassium concentration can lead to severe neuromuscular or muscular dysfunctions [4,5]. Calcium (Ca) plays a crucial role in the formation and maintenance of bone structures and in muscle physiology. It is involved in numerous intracellular signal transduction events and is indispensable for muscle contraction, blood coagulation, hormonal system regulation, nerve impulse transmission as well as energy and fat metabolism [6]. Magnesium (Mg) is indispensable for various enzymatic reactions, protein/nucleic acid synthesis and energy metabolism [7].

It was shown that the prevalence of electrolyte imbalances in the general population may reach up to 15% [8]. Moreover, deviations in electrolyte concentrations can be associated with various diseases such as hypertension [9,10,11,12,13], cardiovascular diseases (CVD) [6,14,15,16,17], osteoporosis [18] and diabetes mellitus type/insulin resistance [19]. Interestingly, major differences in the geographic prevalence of these diseases have been observed in Europe, with higher rates in the eastern and north-eastern, and lower rates in the western and south-western European countries [20,21,22,23]. This “East–West gradient” was also reported for individual European countries including Germany, France and Austria [23,24,25]. The east–west differences within Austria were also found to be apparent regarding food intake, nutrient profile and body composition. The Austrian Nutrition Report 2012 showed that BMI, waist circumference and body fat percentage in adults were significantly higher in eastern Austria compared to western regions. Additionally, east–west differences in energy and fat intake were also reported [26].

In Austria, similarly to many other European countries, tradition plays an important role when it comes to food preferences [27,28]. The western Austrian region is covered by the central Alps; it is less densely populated and predominantly rural. Several traditional foods—such as typical milk products (e.g., Tyrolean cheese), meat and meat products (e.g., Tyrolean Speck), dumplings and sweet dishes (e.g., Kaiserschmarrn, Germknoedel), as well as soups and stews—are protected and registered in the Database of Origin and Registration by the European Commission [29]. They are rooted in Austria’s food culture and, therefore, have an important impact on nutrient intake.

Numerous previous studies have suggested that correlations between specific foods or single nutrients with health outcomes can be inconsistent. Instead, analyses of dietary patterns may better reflect complex eating behaviors and enable correlations to be detected of food preferences with the prevalence of certain diseases. Therefore, it is important to recognize differences in dietary patterns and nutrient profiles within a certain population in order to understand possible associations with regional occurrence of nutritional deficits and diseases.

In our previous work, we identified three dietary patterns that are typical for our geographic area, with more than 60% of population following a traditional dietary pattern. This pattern was characterized by a high intake of animal products, traditional sweet dishes, dumplings, soups and stews with regular use of butter, cream and eggs. The second major dietary pattern in Western Austria, a health-conscious pattern (25% of studied population), was represented by high intake of health-promoting foods such as plant-based products. Only 10.8% of the study population followed a western dietary pattern characterized by low consumption of plant-based products, preferential intake of processed meat, fast food and high-sugar beverages [30].

The aim of the present study was to evaluate the status of electrolyte nutritional intake in Tyrol, western Austria. We further compared electrolyte intake across various age and gender groups and combined data about individual electrolytes with food choices and dietary patterns. Our data relates electrolyte intake to actual food consumption in a tradition-rich region, and provides suggestions for future dietary improvements.

## 2. Materials and Methods

### 2.1. Study Population

The concept, design and selection of the study population has been described before [30]. In brief, over the course of a cross-sectional study, the body composition and eating habits of 463 adults (18–64 years old) in Tyrol (Austria) were assessed between October 2014 and October 2015 by the Department of Dietetics at the Health University of Applied Sciences Tyrol. For a representative study population, Statistics Austria [31] provided the data which was based upon the registry of residents, and stratified by gender and age groups, i.e., 18–24, 25–50 and 51–64 years, as suggested by EFSA [32]. In order to obtain representative results for this region, a minimum net sample of 400 subjects was required with a significance level of ± 5% (*p* < 0.05), within a confidence interval of 95% for a population of 193,386 individuals in the year of 2014. The random sample included 1376 individuals who were provided with a unique identification number. Due to nonresponses, the final number of participants was 358, and thus lower than the required minimum net sample of 400. We also observed a higher response rate in women (32%) than in men (20%). Therefore, additional participants were recruited in regional companies. Exclusion criteria were based on age and permanent place of residence, as well as unusual energy intake (either too low or too high) which was determined by Goldberg cut-off points. As such, a total of 463 subjects (228 men and 235 women) completed the study.

The responsible ethics committee (The Research Committee for Scientific and Ethical Questions), an authorized committee at the UMIT (Private University for Health Sciences, Medical Informatics and Technology), reviewed and approved the study protocol (reference number: 728/13).

### 2.2. Electrolyte Intake

Over the course of one year with consideration of seasonal differences, the survey was conducted on the basis of two nonconsecutive 24-h recall. The first contact consisted of a computer-assisted personal interview (CAPI), and the second of a computer-assisted telephone interview (CATI) six weeks later. By using the computer software Globodiet with the corresponding photo book to obtain the best possible information on food consumption, the participants were asked in detail about the type and quantity of foods consumed in the previous 24-hrecalls. Reconstruction of food consumption was standardized in several phases in order to enhance the precision of memory recollection. Nutritional assessments were planned and carried out in accordance with the best practice guidelines of the EFSA [33] and the German National Consumption Study (NVSII) [34]. Standardized conditions were applied for data collection throughout the whole study. The reported foods collected during the interviews were linked to the German food composition database Bundeslebensmittelschlüssel 3.02 (BLS) [35].

After linking the reported foods to the database, electrolyte intake was determined and compared to the current reference values according to the D-A-CH societies (central European (German (D), Austrian (A), and Swiss (CH); D-A-CH) recommendations) [2] (Table 1). The assessed food items were first categorized into 40 food groups. Those food groups were defined by similarities regarding their nutrient profile (e.g., carbohydrates, dietary fiber, fats, protein) and their affiliation to common main food groups (e.g., milk and milk products, grains, vegetables) based on the classification of the BLS. Based on these food groups, we derived three dietary patterns (health-conscious, traditional and western dietary pattern) by applying a factor analysis of principal components (PCA) followed by a cluster analysis (CA) [30], and compared these patterns with each other regarding electrolyte intake.

Subsequently, for simplification, the forty food groups were further consolidated and reduced to 23 food groups, which contributed most to electrolyte intake. For further analysis, we used absolute sodium, potassium, calcium and magnesium intake values as compared to the D-A-CH references [2]. Additionally, we divided the study population into three groups regarding their level of electrolyte intake (low, normal, high), based on the respective recommendations.

### 2.3. Statistical Analysis

A descriptive analysis of the main characteristics of interest was performed. The full ranges of variation, arithmetic mean, standard deviation, median and interquartile (IQR) range were calculated. To determine group differences, an independent sample *t*-test was applied for gender differences, and a one-sample *t*-test was used for comparison of mean intakes of electrolytes with recommended intake values. One-way ANOVA with Bonferroni’s correction was applied for age and dietary pattern groups, with the former being adjusted for gender and energy, and the latter for gender and age. Multivariate ANOVA was used for analyses of food groups and electrolyte intake. All presented *p*-values are 2-tailed; *p* < 0.05 was considered significant. All analyses were performed using Excel 2016 (Microsoft Corporation, Redmond, Washington) and IBM SPSS Statistics 24 (SPSS Inc., Chicago, IL, USA). The results are presented as tables and box-and-whisker graphs (box plots). The box-and-whisker graphs depicted in the figures represent the absolute median values in mg per day (mg/day); the ends of the whisker are set at 1.5*IQR above the third quartile (Q3) and 1.5*IQR below the first quartile (Q1).

## 3. Results

### 3.1. Assessment of Electrolyte Intake

Subject characteristics are listed in Table 2. We evaluated daily intake levels of sodium, potassium, calcium and magnesium and compared them with the D-A-CH reference values. Absolute median values separately calculated for gender and age groups are depicted in figures (mg/day) and in the Appendix A.

Differences between gender and age groups are shown in Figure 1 and Figure 2, while values adjusted to energy intake are summarized in Table 3. The absolute median values for sodium intake were significantly higher in all persons as compared to the recommended values. Furthermore, men had a higher daily intake of sodium than women (Figure 1, Appendix A). High sodium intake was independent of age, also when adjusted for gender and energy intake (Figure 2, Table 3). Daily potassium intake was significantly lower in all age groups (Figure 1 and Figure 2). There was also a considerable difference between women and men, with women having a higher adjusted intake (Table 3).

Median calcium intake was lower than recommended only in women (Figure 1). However, when adjusted for energy intake, calcium consumption in men and women was equal (Table 3). The median values showed no substantial differences in all age groups (Figure 2). When adjusted for gender and energy intake, calcium intake within the first age group was significantly higher compared to the other age groups (Table 3). The intake of magnesium in different gender and age groups was not different from the recommended values (Figure 1 and Figure 2).

Further, we analyzed dietary intake in more detail. We found that the absolute median sodium intake was highest in the young men group. There was a moderate decrease in sodium consumption with age. Within the first and the third age groups, women had a lower sodium intake compared to men, with no difference in the second age group. When adjusted for energy intake, no significant difference was found (Figure 3a, Table 3).

Importantly, median or mean intake values alone are not sufficient to identify persons that might be affected by mineral intake deficits/excess, and thus, that are at risk of developing electrolyte imbalances. Therefore, we additionally quantified the percentage of persons with a disparity between actual dietary intake and the respective reference values sorted by age groups. We took the reference values as “normal” intake level ± 10% of the error, as previously described [36,37]. Our quantification showed that 75.6–82.9% of individuals had a substantially higher sodium intake (Figure 3b), while only 5.3–6.3% of persons met the daily-recommended values (Appendix A).

Low potassium intake was notable in all age groups, especially in women of all three age groups (Figure 3c). When analyzed as percentage, 74.7–89.9% of individuals had lower potassium intake, with only 5.0–10.7% reaching the recommended values (Figure 3d, Appendix A).

Although median and mean intake values for calcium and magnesium were adequate (Figure 4a,c, Appendix A), additional analysis showed that only 6.7–14.3% of individuals were in the normal range of calcium intake, whereas 26.9–46.7% were in the higher and 42.7–58.8% in the lower range, (Figure 4b). As for magnesium, 40.0 to 45.4% of persons showed adequate intake, 15.1–30.7% were in the higher and 29.3–40.3% in the lower range (Figure 4d, Appendix A).

### 3.2. Main Food Sources of Electrolyte Intake

We analyzed the main food sources for all four electrolytes (Table 4). There were no considerable differences between women and men in all age groups. Bread, grains and potatoes, as well as meat, meat products, fish and condiments were the main sources of sodium. Meanwhile, 59.1% of potassium intake came from nonalcoholic drinks (water and mineral water, tea, fruit and vegetable juice, soda and energy drinks, low sugar beverages, diet soda, coffee and green/black tea), fruits, vegetables, pulses, nuts and seeds, breads, grains and potatoes. Finally, 49.6% of calcium intake came from milk and milk products, while magnesium intake mostly came from bread, grains and potatoes, fruits, vegetables, pulses, nuts and seeds and nonalcoholic drinks.

### 3.3. Electrolyte Intake among Dietary Patterns

We previously derived three dietary patterns in adults in Western Austria and labelled them as “health-conscious”, “western” and “traditional” dietary patterns [30]. In the current study, we analyzed absolute median and mean electrolyte intakes adjusted for gender and age among dietary patterns (Figure 5, Table 5).

Our analysis showed that the absolute median intakes of all four minerals were highest in the western dietary pattern group. Median sodium and calcium intakes were lowest in the health-conscious dietary pattern group, while potassium intake was lowest in individuals following the traditional dietary pattern. When looking at the adjusted mean electrolyte intake (Table 5), sodium intake was lowest within the health conscious dietary pattern group, compared to the traditional and the western dietary pattern groups. Adjusted mean potassium and magnesium intake was lowest among individuals following the traditional dietary pattern, while calcium intake was lowest among those following the health-conscious dietary pattern. Regarding all electrolytes, we observed significant differences between dietary pattern groups when comparing adjusted values, with the exception of calcium.

We further quantified persons that reached the reference levels of all four electrolytes (D-A-CH values +/− 10% of error, as described before); we found that only 5 persons (1%) in our study population did so. A further 60 persons (13%) consumed adequate levels of three out of four electrolytes. Moreover, 66 persons (14%) did not reach reference levels for any of the analyzed electrolytes. We further analyzed these two “polar” groups in order to identify which food groups were responsible for the observed differences (Table 6). As shown in Table 6, the prominent differences between “electrolyte-adequate” Group 2 and Group 1 were: (1) water/mineral water increase along with decrease of sodas/energy drinks; (2) high consumption of fruits/vegetables/legumes and milk/cheese products (3) decreased consumption of white bread; (4) increased consumption of vegetable oils/nuts/seeds along with decreased butter/margarine and (5) low consumption of sweets/cakes. Due to the low number of persons in both groups, we could not reliably correlate adequate electrolyte intake and previously defined dietary patterns (data not shown).

## 4. Discussion

The present study is the first combined analysis of dietary sodium, potassium, calcium and magnesium intake in Austria. We compared nutritional intake of these four electrolytes in different gender/age groups, and described the main food sources of electrolytes according to regional dietary patterns.

It was previously shown that western dietary practices with a higher consumption of cereals and low-nutrient-density processed foods, and a lower consumption of fruits and vegetables, led to a diet high in sodium and low in potassium [38,39,40]. Additionally, 99.4% of US adults were reported to consume excessive sodium, whereas less than 2% attained the recommended values for potassium [39]. Additionally, meat products (including processed meats such as ham, bacon, etc.) contribute to a high proportion of salt intake worldwide [41]. The reported potassium intakes below estimated average requirement (EAR) were 11–63% for women and 3–37% for men, and were highest in Spain, France, Poland and the UK, and lowest in The Netherlands [42]. Deficits in magnesium intake in the adult population (18–60 years) were also previously reported in France (32% below EAR) and the UK (36% below EAR).

Our results confirm and extend previous studies in Austria, Germany, the UK and the US, where 50% or more of the participants did not reach the recommendations [34,36,42,43,44]. In all three dietary patterns previously defined in Tyrol, the mean value of sodium intake was higher than the suggested consumption, which is also in line with our previous studies showing high consumption of salt in all regions of Austria [44,45]. Analyses of potassium, calcium and magnesium intake showed that the majority of our study population did not reach the normal intake range. When combining electrolyte intake values, only 1% of our study population reached D-A-CH recommended levels [2], while 13% consumed adequate levels of three electrolytes. Analyses of food groups showed that these individuals preferred water/mineral water over sodas/energy drinks, consumed high amounts of fruits/vegetables/legumes and milk/cheese products, preferred vegetable oils over butter and at less sweets/cakes. Due to the low number of persons, we could not reliably correlate adequate electrolyte intake with previously defined dietary patterns, but our data may help to correct existing patterns by focusing on defined food groups.

Our data may extend the discussion about the practical applicability of reference values for individual electrolytes. Thus, the optimum calcium intake remains uncertain. According to the Austrian dietary guidelines [46], three servings (about 500 g/day) of milk and dairy products (e.g., 1 glass of milk, 1 cup of yogurt, one piece of cheese) are recommended with a focus on covering calcium requirements. Guidelines from other geographic regions reduced their reference values for ecological reasons and taking into account plant-based calcium sources such as green vegetables, nuts and seeds, mineral water rich in calcium or milk alternatives [47]. Although milk and dairy products often come under criticism by the media and the public, the current scientific literature suggests that the consumption of an appropriate amount of milk and dairy products may be beneficial at all ages, with the exception of individuals with specific medical conditions such as lactose intolerance or milk protein allergy [48]. A recent review by Thorning et al. [49] summarized data on the benefits of milk products, suggesting a protective effect against type 2 diabetes and CVD, and a positive effect on bone mineral density with very few adverse effects. At the same time, studies from Asian regions report calcium intakes of less than 500 mg/day [50], and suggested that low intake of calcium is not associated with a higher risk of bone fractures, diabetes mellitus type 2, CVD [51,52] and hypertension [35]. A recent publication by Willet et al. [53] reviewed global healthy diets and suggested that among adults, the risk of fractures is not substantially reduced with calcium intakes greater than 500 mg/day. A clear association in this matter is difficult to assess, since other factors (e.g., genetics, calcium absorption, intolerances, etc.) have to be taken into account [54]. However, higher intake of calcium or milk was not associated with CVD and other diseases [17], which makes it even more difficult to establish connections between calcium intake and the aforementioned health issues. With this research background, our study population seems to be at low risk of calcium deficiency, despite their relatively low intake of milk and dairy products.

Evidence-based potassium recommendations have also been widely discussed in the literature. Current recommendations differ between China, the US and Europe, and are suggested to be considerably higher than current average intakes [55]. The D-A-CH reference value for potassium was recently changed from 2000 mg/day to 4000 mg/day, which makes it more difficult to achieve [2]. Thus, the large gap between recommended intake and the actual consumption of potassium, as we and other researchers have observed, has to be further evaluated, especially in the context of the whole diet including energy values and other minerals.

Inadequate magnesium status was previously observed in populations which consume processed-foods [43]. Our data show that the magnesium intake mean value does not differ significantly from recommended intake levels. Still, 40% of our study population did not consume enough magnesium. Most of this mineral in our study came from bread, grains and potatoes (24.4%), fruits, vegetables, pulses, nuts and seeds (18.7%) and nonalcoholic drinks (18.4%). Interestingly, the average intake of whole grains, fruits, vegetables, legumes, nuts and seeds was still significantly lower than the nutritional guidelines suggest. According to the Austrian dietary guidelines [31], three portions of vegetables (375–600 g/day) and two portions of fruits (250–300 g/day) should be consumed daily. In our study population, women only reached 36.9% (1.1 portions) of the recommended vegetable and 61.0% (1.22 portions) of the recommended fruit intake, while men reached 37.3% (1.1 portions) of the recommended vegetable and 43.9% (0.9 portions) of the recommended fruit intake (17). Although magnesium mean value intake in our study population was mostly within the recommended ranges, there was a clear need to optimize overall electrolyte intake by incorporating more fruits and (green) vegetables, legumes such as (soy) beans, peas, lenses and chick peas as well as whole grains such as whole grain bread, oats, or whole grain pasta. In addition, nuts and seeds such as pumpkin, sesame and sunflower seeds, as well as nuts, contribute to an adequate electrolyte intake. Taking into account that most participants follow a traditional dietary pattern, which does not include most of these products, diet optimization is challenging.

The results of our analysis allow several applicable conclusions to be made concerning the local population. Persons following traditional dietary patterns consume enough calcium and magnesium, but also eat too much salt and significantly less potassium, compared to the other dietary patterns in our study. It is therefore recommended that the consumption of plant-based foods that are widely accepted in Tyrolean cuisine be increased, e.g., white and red cabbage, potatoes, sauerkraut (pickled cabbage), seasonal local fruits and vegetables including spinach, leeks, radish, asparagus, broccoli, root vegetables, pears and apples. Persons in the group of the health-conscious dietary pattern had better sodium and potassium intakes, but did not reach the recommended calcium levels. Individuals following this dietary pattern often consume foods that are plant-based [30]; thus, if it is not possible to increase calcium intake by consuming more dairy products, alternative products, such as fortified, plant-based milk, should be included.

*Strengths and Limitations:* The use of two nonconsecutive 24-h recalls is one of this study’s strengths, since we could consider seasonal differences and achieve the requirements of standardized conditions, as recommended by the EFSA [17]. However, the results were limited by the cross-sectional design of the study, which only allowed judgements to be made regarding possible associations but not of casual relations. Also, we could only recruit participants from the central area of Tyrol, and therefore, the results are only representative of that area. Further limitations regarding our study design are described elsewhere [14]. With lower response rates in men than in women, nonresponse bias is likely, and it was necessary to additionally recruit participants from regional companies to complete the missing gender and age groups. Therefore, these persons were not part of the original random sample. Furthermore, although known confounders were considered, residual confounding remains a possibility. In this study, the “migration background” of the person was considered when at least one of the parents was not born in Austria (23.5% of participants). We did not observe any significant differences in BMI or health parameters in this group of participants [56].

Furthermore, dietary surveys are often not considered optimal for estimating a population’s electrolyte intake, due to the variability of electrolyte content in the food and the use of 24-h recalls. Additionally, no blood and urine samples were taken, and thus, actual electrolyte status could not be assessed.

## 5. Conclusions

Our results add new data regarding the low potassium intake of all gender and age groups investigated in our study. We also confirm the previously described urgent need to decrease salt intake. When compared to D-A-CH recommendations, none of the dietary patterns in Tyrol was optimal for sodium, potassium, calcium or magnesium intake. Considering that more than 60% of our study participants follows a traditional dietary pattern, it is important to optimize this diet and provide adequate alternatives in food choice and/or preparation methods. Together with previously published data, our new results suggest increasing consumption of regional traditional fruits, vegetables and legumes. Further modifications of traditional cooking methods, such as replacing butter with vegetable oils when frying or sautéing foods, will clearly be beneficial. For persons following a health-conscious dietary pattern, it is important to consume more calcium-rich foods such as milk products and cheese, or to find alternative calcium sources. Further investigations are needed to develop dietary strategies which are applicable to tradition-reach geographic areas.

## Figures and Tables

**Figure 1 nutrients-12-01956-f001:**
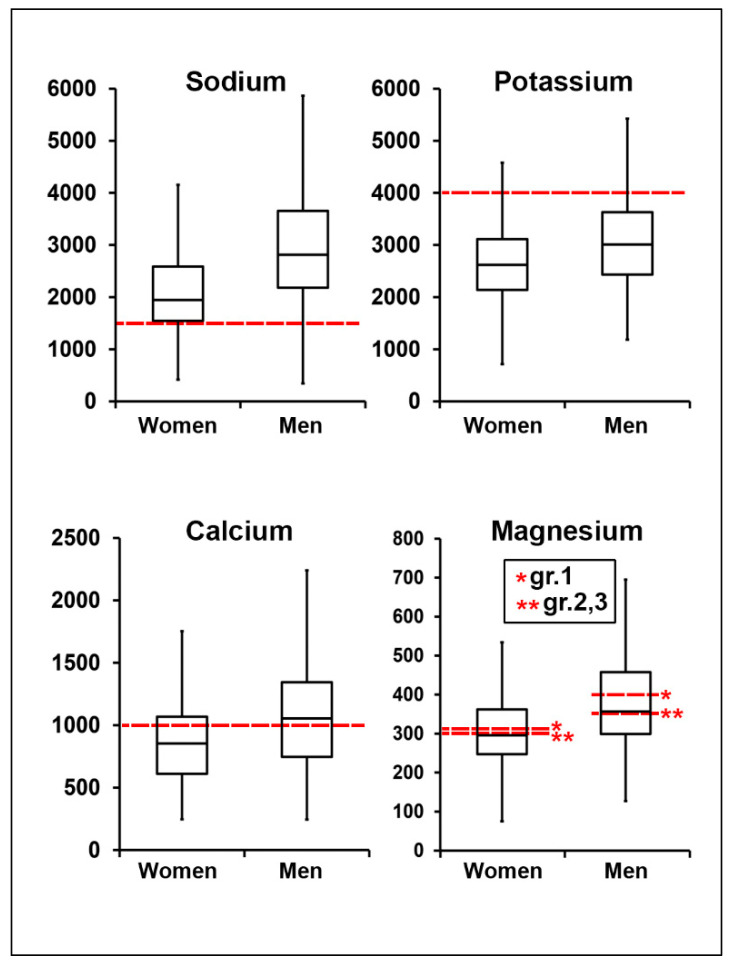
Median intake levels of sodium, potassium, calcium and magnesium in women and men, mg/day. Red dashed lines depict recommended intake levels for each electrolyte. For magnesium, red dashed lines depict reference values for different age groups: group 1 (one asterisk *gr.1) and group 2, 3 (two asterisks **gr.2, 3) accordingly.

**Figure 2 nutrients-12-01956-f002:**
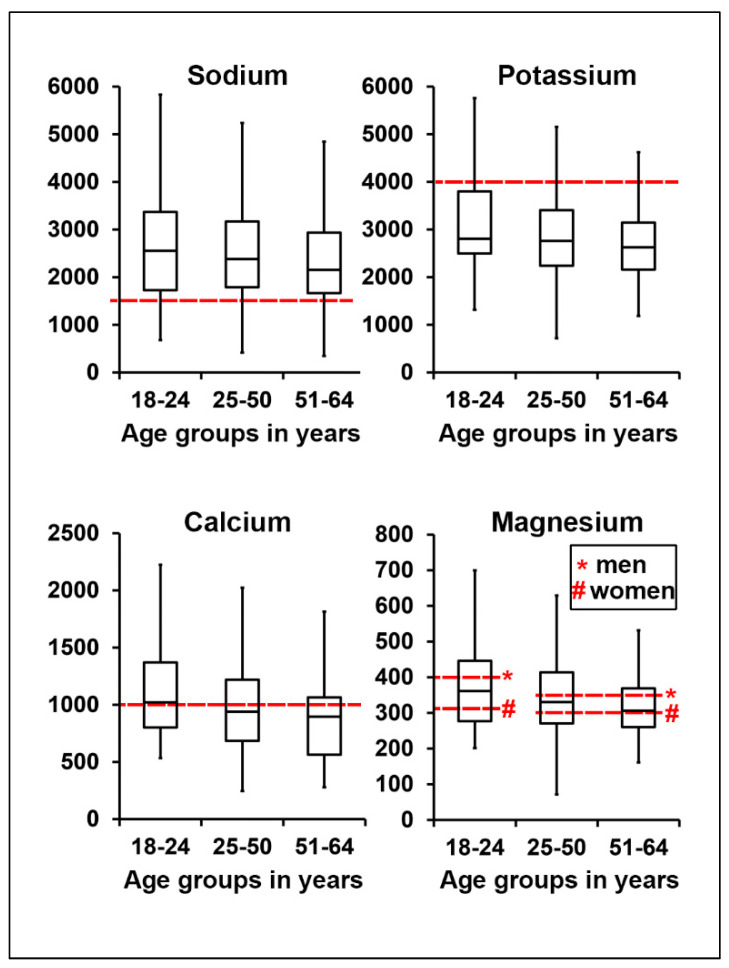
Median intake levels of sodium, potassium, calcium and magnesium in different age groups in mg/day. Red dashed lines depict recommended intake levels for each electrolyte. For magnesium, red dashed lines depict reference values for different gender groups: men (asterisk *) and women (hash sign #) accordingly.

**Figure 3 nutrients-12-01956-f003:**
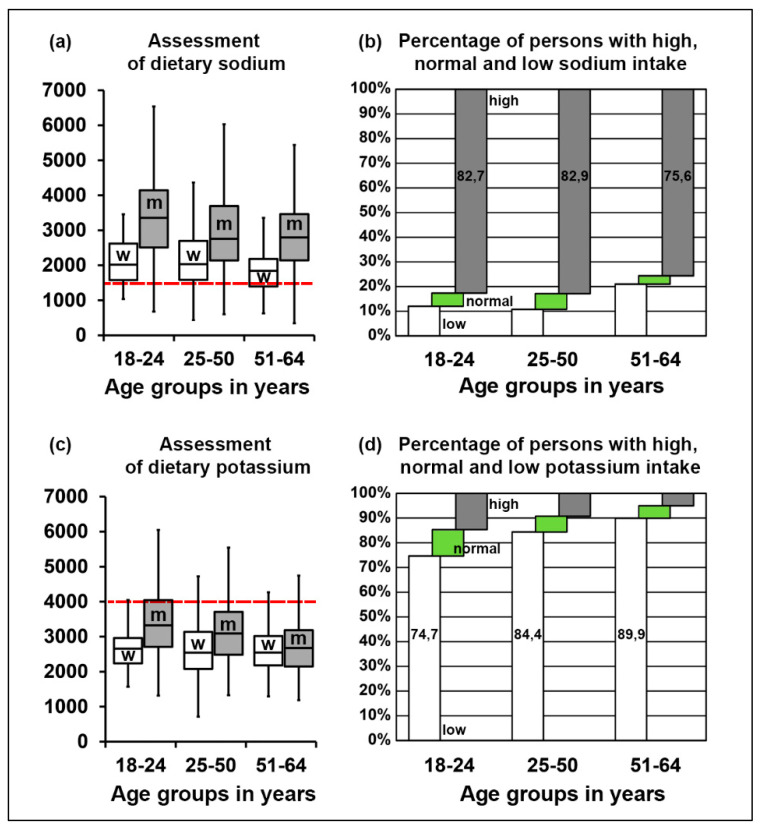
(**a**) Median intake levels of sodium in different gender and age groups in mg/day. (**b**) Percentage of persons with a discrepancy between reported sodium intake and the reference values in different age groups. (**c**) Median intake levels of potassium in different gender and age groups in mg/day. Red dashed lines depict the recommended intake level for sodium (**a**) and potassium (**b**); w indicates women, m–men. (**d**) Percentage of persons with a discrepancy between reported potassium intake and reference values in different age groups. See also Appendix A.

**Figure 4 nutrients-12-01956-f004:**
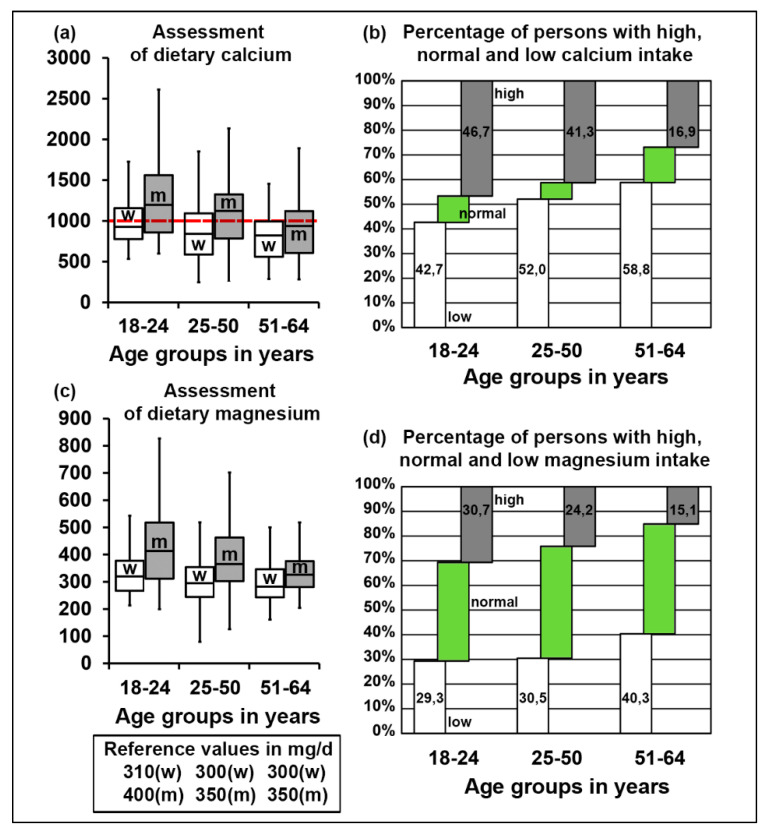
(**a**) Median intake levels of calcium in different gender and age groups in mg/day. Red dashed line depicts the recommended intake level for calcium, w indicates women, m–men. (**b**) Percentage of persons with a discrepancy between reported calcium intake and the reference values in different age groups. (**c**) Median intake levels of magnesium in different gender and age groups in mg/day. Reference values for different gender and age groups are specified in the table below, w indicates women, m–men. (**d**) Percentage of persons with a discrepancy between reported magnesium intake and the average of reference values in different age groups. See also Appendix A.

**Figure 5 nutrients-12-01956-f005:**
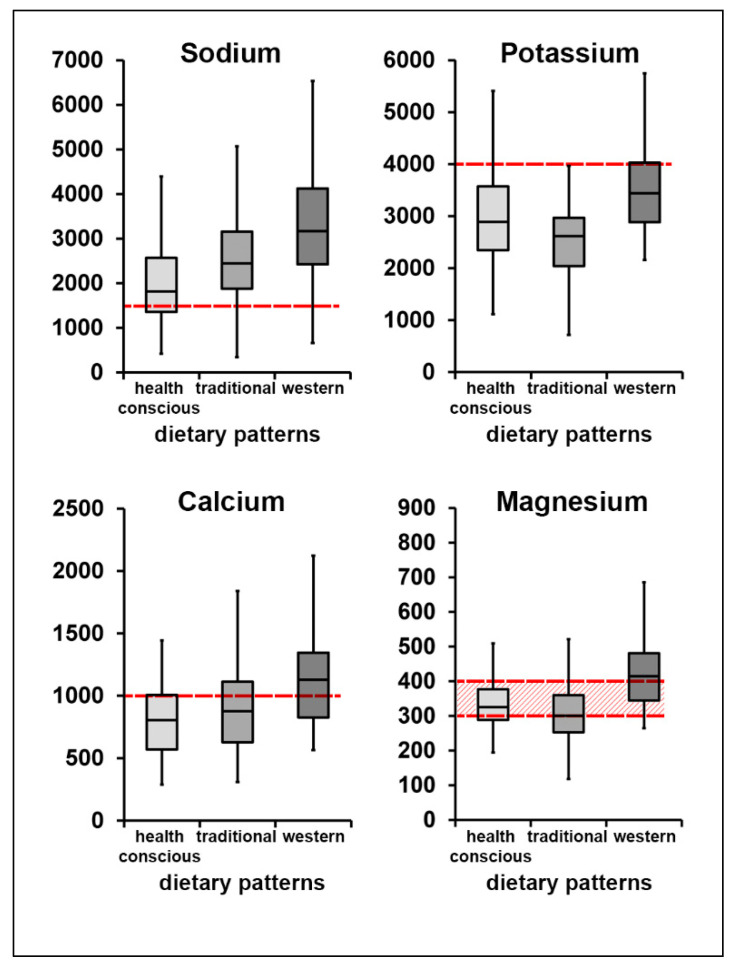
Median intake levels of electrolytes in different dietary patterns in mg/day. Red dashed lines depict recommended intake levels for each electrolyte. For magnesium, the red dashed pattern shows distribution of reference values for different gender/age groups.

**Table 1 nutrients-12-01956-t001:** Reference Values for electrolyte intake in mg/d according to the Nutrition Societies of Germany, Austria and Switzerland (D-A-CH) [2].

Electrolytes	Women	Men
Sodium	1500	1500
Potassium	4000	4000
Calcium	1000	1000
Magnesium		
age 18–24 years	310	400
age 25–64 years	300	350

**Table 2 nutrients-12-01956-t002:** Characteristics of the study population.

Characteristics	Women	Men	Total
*n*	235	228	463
Age, years *	41.0 (12.9)	39.9 (12.9)	40.4 (12.9)
Height, cm	166.7 (5.9)	178.9 (7.2)	172.7 (8.9)
Weight, kg	65.5 (13.4)	81.4 (14.6)	73.3 (16.1)
Waist circumference, cm	85.9 (11.71)	91.39 (11.8)	88.6 (12.9)
Hip circumference, cm	100.2 (10.1)	100.9 (7.4)	100.5 (8.9)
Waist-to-hip ratio	0.86 (0.06)	0.90 (0.07)	0.88 (0.07)
BMI, kg/m^2^	23.6 (4.9)	25.4 (4.0)	24.5 (4.5)
Overweight/obese, %	28.9	45.6	37.3
Smoking, %	21.7	26.8	24.3
Migration background **, %	23.8	23.2	23.5
Education, %			
Elementary school	0.4	0.0	0.2
Secondary school	0.9	0.9	0.9
Apprenticeship	14.6	26.6	20.4
Professional school	17.2	9.0	13.2
School with higher education (Matura)	19.7	24.3	22.0
University	34.8	31.1	33.0
Other	12.4	8.1	10.3
Occupation, %			
Full-time	63.9	86.8	75.8
Part-time	36.1	13.2	24.2

* all metric variables are means with standard deviation (SD) ** one or both parents were not born in Austria.

**Table 3 nutrients-12-01956-t003:** Electrolyte intake in association with gender and age groups. Data shown as adjusted mean values and standard error of the mean (SE) in parentheses in mg/day.

Characteristics	Sodium	*p*-Value ^1^	Potassium	*p*-Value	Calcium	*p*-Value	Magnesium	*p*-Value
gender ***								
women	2412.0 (58.9)		3041.8 (63.5)		983.6 (23.3)		354.1 (7.6)	
men	2639.8 (59.9)	0.011	2824.7 (64.6)	0.024	977.5 (23.7)	0.862	352.5 (7.7)	0.887
age groups **		0.31		0.658		0.034		0.319
18–24 years	2487.9 (98.9)		2922.9 (106.1)		1048.1 (38.8)		365.5 (12.7)	
25–50 years	2575.7 (51.6)		2961.7 (55.3)		987.7 (20.3)		355.0 (6.6)	
51–64 years	2438.7 (77.9)		2870.4 (83.6)		921.0 (30.6)		341.4 (10.0)	
women *		0.083		0.335		0.18		0.184
18–24 years	2044.7 (107.9)		2760.8 (131.3)		946.1 (46.4)		339.7 (15.7)	
25–50 years	2159.9 (57.9)		2648.0 (70.5)		860.6 (24.9)		308.8 (8.4)	
51–64 years	1928.8 (86.5)		2829.0 (105.2)		840.5 (37.2)		325.1 (12.6)	
men *		0.899		0.06		0.105		0.072
18–24 years	2926.8 (169.4)		3090.6 (169.3)		1145.0 (63.2)		393.1 (20.3)	
25–50 years	3005.6 (85.8)		3286.5 (85.7)		1120.3 (32.0)		401.8 (10.2)	
51–64 years	2959.5 (131.3)		2922.2 (131.2)		1005.3 (49.0)		358.8 (15.7)	

^1^*p*-values for group differences are based on ANOVA for metric variables *** adjusted for energy intake. ** adjusted for gender and energy intake.

**Table 4 nutrients-12-01956-t004:** Main food groups contributing to electrolyte intake (shown in %).

Main Food Groups	Sodium	Potassium	Calcium	Magnesium
Water and mineral water, unsweetened tea	1.4	0.9	12.5	6.4
Fruit and vegetable juice	0.1	3.8	1.0	2.0
Sodas and energy drinks	0.5	0.2	1.3	1.1
Low sugar beverages, diet soda	0.1	0.9	0.3	0.4
Coffee and green/black tea	0.6	8.0	3.7	8.5
Fruits, vegetables and legumes	5.8	26.3	7.5	14.5
White bread	13.0	2.8	3.2	5.7
Whole grain bread	11.4	5.0	1.8	8.1
Refined grains	5.7	3.0	2.2	5.8
Whole grains	0.3	1.4	0.4	3.8
Potatoes	0.6	6.2	0.7	3.0
Milk and milk products	3.1	8.2	19.9	5.7
Cheese	9.6	2.2	29.7	3.3
Vegetable oils, nuts and seeds	0.0	1.8	0.8	4.2
Butter/margarine	0.1	0.1	0.1	0.1
Eggs	0.1	0.0	0.0	0.0
Meat and meat products	11.5	7.8	1.2	5.3
Fish and fish products	8.9	2.6	0.6	2.2
Sweets, desserts and cakes	2.5	7.8	5.8	9.0
Salty snacks and fast food	1.5	2.1	0.5	1.4
Alcoholic beverages	0.3	2.8	1.0	4.4
Condiments, sauces, spices, artificial sweeteners	21.6	3.6	3.5	3.7
Others (soy products, plant-based milk, soups, protein drinks, beer/wine alcohol-free)	0.3	2.1	1.2	2.4

**Table 5 nutrients-12-01956-t005:** Electrolyte intake among dietary patterns adjusted for gender and age.

Characteristics and Electrolyte Intake	Health-conscious Dietary Pattern	Traditional Dietary Pattern	Western Dietary Pattern	*p*-Value
	*n* = 118	*n* = 295	*n* = 50	
Gender, % of women	72.9	48.8	10.0	<0.001
Age, year *	39.8 (1.1)	42.5 (0.7)	29.9 (1.8)	<0.001
Energy intake, kcal/day **	1901.5 (61.7)	2161.7 (35.0)	2445.2 (141.2)	<0.001
Electrolyte intake **				
Sodium, mg/day	2113.1 (106.3)	2594.9 (60.3)	2899.1 (243.2)	<0.001
Potassium, mg/day	3289.4 (108.2)	2688.8 (61.4)	3285.5 (247.6)	<0.001
Calcium, mg/day	939.7 (40.2)	970.2 (22.8)	1091.1 (91.9)	0.317
Magnesium, mg/day	410.3 (12.5)	321.8 (7.1)	393.5 (28.6)	<0.001

* Values are means and standard error of the mean in parentheses. ** adjusted for gender and age.

**Table 6 nutrients-12-01956-t006:** Comparison of main food groups contributing to electrolyte intake in participants with adequate electrolyte intake of at least three electrolytes with persons that did not reach reference values in all electrolytes.

Main Food Groups	Participants		*p*-Value
	Group 1 (*n* = 66)	Group 2 (*n* = 65)	
Water and mineral water, unsweetened tea	1410.2 ± 138.6	1952.5 ± 131.1	0.002
Sodas and energy drinks	423.5 ± 54.5	100.8 ± 51.6	0.002
Fruits and vegetables and legumes	199.5 ± 37.4	508.3 ± 35.4	<0.001
White bread	86.3 ±9.0	32.6 ± 8.4	0.001
Milk and milk products	70.4 ± 25.0	284.2 ± 23.6	<0.001
Cheese	32.8 ± 11.7	97.6 ± 11.0	0.004
Vegetable oils. nuts and seeds	7.7 ± 3.3	24.3 ± 3.1	0.046
Butter/margarine	12.8 ± 1.6	8.0± 1.5	0.013
Sweets. desserts and cakes	149.00 ± 16.3	80.5 ± 15.4	0.034
Others (soy products, plant-based milk, soups, protein drinks, beer/wine alcohol-free)	72.1 ± 28.0	208.7 ± 26.5	0.022

Multivariate ANOVA model, adjusted by gender, age and energy. Values are depicted as mean value ± standard error of the mean in mg/day. Group 1: persons who did not reach reference values for any of the four electrolytes. Group 2: persons that reached reference values for at least three electrolytes.

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
