# Peer review of "Electrolyte Intake and Major Food Sources of Sodium, Potassium, Calcium and Magnesium among a Population in Western Austria"

_nutrients, 2020, doi:10.3390/nu12071956_

Round 1
Reviewer 1 Report
In my opinion the scietific value of this manuscript is rather low. What is the novelty of this study? The study could contain the element of novelty if it will be focused on the three dietary patterns : health, traditional, western and also include more nutritional and health parameters and validated methods. Discussion is too general and it should be more related to the obtained results; in the discussion Authors should try to explain the obtained significant changes. Conclusion is general and also includes well known information.
Author Response
Reviewer 1
Comment 1
In my opinion the scietific value of this manuscript is rather low. What is the novelty of this study?
The study could contain the element of novelty if it will be focused on the three dietary patterns: health, traditional, western and also include more nutritional and health parameters and validated methods.
Answer
Following suggestions of Reviewer 1 and Reviewer 2 we have performed a major revision of our manuscript and added more data on analysis of dietary patterns in addition to individual electrolytes.
Major changes include:
- We have refocused our revised manuscript on dietary patterns in addition to individual electrolytes. Taking in account significant amounts of scientific literature concerning associations between individual electrolytes and health outcomes, we decided to leave our originally performed detailed analysis as an important part of the manuscript and combine it with new data analysis.
- We have additionally analyzed food choices to identify a dietary pattern that covers all electrolytes intake requirements according to reference values of our geographical region (D-A-CH values). Only 1% of our study population (5 persons) could fulfil requirements for all four electrolytes and further 60 persons could reach suggested values for three out of four electrolytes. 66 persons did not reach any reference values for any of four electrolytes analyzed in our manuscript. We have further compared food choices of these two “polar” groups and added Table 6 (page 14) that summarizes the results. Due to the low number of individuals in these two groups we could not confirm one dietary pattern that would reliably apply.
Our results are of special interest and follow current discussions of relevant and practical applicability of singular reference values, especially for potassium that was recently changed from 2000mg/d to 4000mg/d in the D-A-CH guidelines. Our results also support the current research view on the importance of health beneficial dietary patterns rather than focusing on individual nutrients and reference values.
To the best of our knowledge, our study is the first study that combines detailed analysis of dietary sodium, potassium, calcium and magnesium intake in Austria, compares nutritional intake of these four electrolytes in different gender/age groups; and describes electrolyte intake within regional dietary patterns.
- We have revised Introduction, Results and Discussion parts according to the Reviewers’ suggestions and included a discussion of new data. The newly added text is highlighted in green.
Comment 3
Discussion is too general and it should be more related to the obtained results; in the discussion Authors should try to explain the obtained significant changes.
Answer
We have performed major revision of the Discussion part and focus on explanation of the obtained significant changes.
Comment 4
Conclusion is general and also includes well known information.
Answer
We have revised Conclusion part that is now more focused on our findings
Reviewer 2 Report
Comments and suggestions attached

Author Response
Reviewer 2
We have revised our manuscript according to suggestions& comments of Reviewer 2. Newly added Table and text are highlighted in green.
Specific comments:
Comment 1
If I understand correctly, of 1376 randomly selected only 463, 34% qualified. It begs the question of how 1 in 3 can be representative. Perhaps clarification here would be helpful.
Answer
In order to obtain representative results for this region, a minimum net sample of 400 subjects was required. We calculated with a respondents rate of 30-35%, so the the random sample out of 193,386 residents in the year of 2014 included 1 376 individuals who were provided with a unique identification number. Due to a higher non-response-rate than calculated, the eventual number of participants was 358 and thus lower than the required minimum net sample of 400. We also observed a higher response rate in women (32 %) than in men (20 %). Therefore, additional participants were recruited in regional companies. Finally, a total of 463 subjects (228 men and 235 women) completed the study. Following this comment of Reviewer 2 we have added an additional explanation in the “Material and Methods” section (page 3, Lines 103-106).
Comment 2
This is not clear – is this the proportion of immigrants? If so, how does it bias, or not, the “traditional Tyrolean” diet, or the diet categories. It is almost a ¼ th of the sample. Are the diets represented similarly in this group, how homogenous is it in terms of diet, or/and does it represent a different immigrant diet? It would be important to clarify these issues.
If there is a different pattern it should appear in tables and supplementary data. If not, this should be mentioned, and preferably supported statistically.
Answer
The population in Tyrol is ethnically more homogeneous than in the Central and Eastern Austrian regions (for example Vienna). Previous studies underlined a strong influence of migration background on dietary patterns, energy intake, BMI and other parameters. In our data set, we have defined each person where at least one of the parents was not born in Austria as “migration background” to better describe the study population. We did not find significant differences in energy intake and BMI (Reference: Egg, S.; Erler, J.; Purtscher, AE. 1. Tiroler Ernährungsbericht. Hg. v. FH-Studiengang Diaetologie. fhg -Zentrum für Gesundheitsberufe Tirol GmbH )
We agree with the Reviewer, that this is an important issue and we can not completely exclude the influence of migrant background on dietary patterns. Therefore, we added this information in “Strengths and Limitations” section (page 16, Lines 369-371): “In this study population migration background was considered when one of parents was not born in Austria (23.5 % of participants). We did not observe any significant differences in BMI or health parameters in this group of participants”. We also added explanation in Table 2 (page 5): “**one or both parents were not born in Austria”.
Comment 3
“female” is a biological term used for animals. While strictly humans too are animals, I would consider it more respectful toward your participants to term them ‘women’ and ‘men’.
Answer
We have corrected term “female” to term “women” and “men” throughout the revised manuscript.
Comment 4
Looks like significantly lower electrolyte intake. Is there a dietary pattern that may account for this? Body weight?
Answer
We did not find a significant difference when values were adjusted to energy intake. Most likely, the deficit may come from decreased calorie intake.
Comment 5
Consider whether figures 3,4 & 5 may be redundant given the additional descriptions in text and table 4
Answer
We agree with Reviewer 2 that Figures 3&4 and Table 4 may be redundant while representing the same data set. However, we find that graphical presentation brings more value to the manuscript. We therefore moved Table 4 into Supplementary material (new “Supplementary Table 2, page 19).
Comment 6
Table 6 is cutoff at both ends in my PDF viewer. I cannot see all the data and headings.
Answer
We apologize for the inconvenience, Table shift might have happened during PDF re-formatting.
Comment 7
In accordance with the primary aim of this paper, ie to identify dietary pattern of electrolyte intake, I would think that instead of, or in addition to using the previously derived dietary patterns and considering their contribution to electrolyte intake, for this article, deriving dietary patterns from electrolyte intake would be informative. For example, what dietary pattern would provide the optimal electrolyte intake, rather than individual electrolytes, and how does the population sample distribute accordingly. Indeed, if such a dietary pattern emerges, it might be more translational in terms of dietary advice than the 3 patterns used here. It might also be more relevant to correlating with disease, one of the stated aims of this research (cf Abstract).
I would add that current views tend to stress individual nutrients less than nutrient interactions, accordingly it might present a more nuanced view to consider overall electrolyte intake as a dietary concern rather than individual electrolytes for both diet and disease.
Answer
This comment of Reviewer 2 is very well taken and we believe that the suggestions of Reviewer 1 and Reviewer 2 to add additional data about dietary patterns strongly improved and enhanced conclusions of our manuscript.
We have made following changes into the revised version:
- We have refocused our revised manuscript on dietary patterns in addition to individual electrolytes. Taking in account significant amounts of scientific literature concerning associations between individual electrolytes and health outcomes, we decided to leave our originally performed detailed analysis as an important part of the manuscript and combined it with new data analysis.
- We have additionally analyzed food choices to identify a dietary pattern that covers all electrolytes intake requirements according to reference values of our geographical region (D-A-CH values). Only 1% of our study population (5 persons) could fulfil requirements for all four electrolytes and further 60 persons could reach the suggested values for three out of four electrolytes. 66 persons did not reach any reference values for any of four electrolytes analyzed in our manuscript. We have further compared food choices of these two “polar” groups and added Table 6 (page 14) that summarizes results. Due to low number of individuals in these 2 groups we could not confirm one dietary pattern that would reliably apply.
Comment 8
Do you mean that 95% of Europeans had adequate K+?
Answer
This comment of Reviewer 2 is applied to the sentence “In most European countries the prevalence of low potassium intake was 5% and ranged from 0 to 31% with the lowest intake in Poland, France and the UK (Mensink GBM et al 2013, PMID: 23312136).” from our original submission.
It has slipped our attention that 5% in this publication is referred to LRNI (lower reference nutrient) that was described in addition to estimated average requirement (EAR). The latter is more relevant for our publication than LRNI, therefore we changed this sentence in revised version to “The reported potassium intakes below EAR were 11–63% for women and 3–37 % for men and were highest in Spain, France, Poland and the UK and lowest in The Netherlands (Mensink GBM et al 2013, PMID: 23312136)”.(page 14, Lines 281-283)
Comment 9
“non-alcoholic drinks” Please clarify what these are (for an international audience). Coffee? Juice (fresh?), soft drinks, milk?
Answer
This point of Reviewer 2 is well taken. We have added the items defining “non-alcoholic drinks” on Page 10 (Lines 229-230): “…non-alcoholic drinks (water and mineral water, tea, fruit and vegetable juice, soda and energy drinks, low sugar beverages, diet soda, coffee and green/black tea)…”
Comment 10
And sodium? Does it counter acidosis too? If so, shouldn’t this be mentioned and considered, even if not at high levels for a healthy diet? For example, is it possible that the high Na+ reduced the acidosis in this dietary pattern, and reducing Na+ would increase it – so that some replacement strategy should be considered?
Answer
We have re-structured our manuscript and re-wrote Discussion, where this section was removed due to different focus.
Comment 11
My suggestion is more one of emphasis than major change. It is that the article vacillates between considering each electrolyte separately, and suggesting both individual electrolyte dietary analysis and advice on the one hand, and very general advice for a healthy plant-emphasising diet on the other.
I believe the article would be more useful if, based on their data, authors could a) put together a diet that would adequately provide all the electrolytes and b) tailor a supplementary diet that would meet the requirements of their specific Tyrolean sample/population. Perhaps even using available Tyrolean foods and to compose a superior diet. This might be more useful that a general recommendation to eat more veges, less sodium etc.
Answer
We followed the suggestions of Reviewer 2 and revised our manuscript accordingly. We now focus on local foods and added details about Tyrolean traditional and local products. We specifically discuss how dietary patterns can be adjusted to cover for electrolytes and give suggestions about local plant-derived products that are popular in our geographical region.
Round 2
Reviewer 1 Report
Authors have corrected the manuscript according to my comments. In my opinion in this form the paper may be published.